# The Climb Carves Wisdom Deeper Than the Summit: On the Importance of Reasoning Patterns

## Abstract

Unlike typical RL studies on verifiable tasks like math, we investigate the more practical challenge of noisy rewards from non-verifiable, real-world tasks. We begin by artificially injecting noise (flipping rewards) into verifiable tasks (e.g., math and question answering) to gain some insights. Surprisingly, we found that rewarding a large portion of outputs with incorrect answers does not hinder the acquisition of effective reasoning abilities. Thus, we hypothesize that the reasoning process itself must be valuable. We validate this hypothesis with a simple yet effective mechanism: the Reasoning Pattern Reward (RPR), which rewards only the appearance of key reasoning phrases (e.g., "first, I need to") without verifying answer correctness. Using RPR, the model achieves peak downstream performance comparable to that of models trained with clean, verified rewards. Recognizing the importance of the reasoning process, we developed a core method that uses RPR to calibrate noisy reward models in open-ended NLP tasks. By incorporating RPR, we effectively mitigate potential false negatives in reward signals, thereby enhancing the LLM's reasoning capabilities and evaluation performance on such tasks. Our findings are validated across both Qwen and Llama model series. These findings provide new insights for advancing post-training techniques.[1]

## 1 Introduction

Reinforcement learning (RL) applied to post-training large language models (LLMs) has led to significant advancements in enhancing their thinking and reasoning abilities (DeepSeek-AI, 2025; Team, 2025), resulting in improved performance on many challenging downstream tasks. Most current research focuses on math tasks (Hu et al., 2025; Pan et al., 2025; Yeo et al., 2025; Gandhi et al., 2025; Chen et al., 2025), as these can be easily verified as correct or incorrect by simple rule-based reward functions. However, in many real-world applications, such as preference alignment (Ouyang et al., 2022; Zhu et al., 2023) and open question-answering (Jaques et al., 2020; Nakano et al., 2022), responses cannot be easily quantified with simple rule-based functions and instead require evaluation by neural reward models. These models, being imperfect, often introduce noise even resulting in opposite rewards. In this paper, we study such noisy and open-domain conditions, with the goal of providing practical insights for developing more robust and general reasoning models suited for real-world deployment.

In Section 3, we begin with verifiable tasks such as math and multi-choice QA, due to their controllable and quantifiable rewards, which provide an ideal testbed for obtaining preliminary insights. We introduce noise by flipping rewards. For example, we assign a reward of 0 for a correct answer and 1 for an incorrect one. Surprisingly, even under substantial noise, such as flipping 40% of rewards to incorrect values, a Qwen-2.5-7B model (Yang et al., 2024) improved its accuracy on MATH-500 (Hendrycks et al., 2021) to 72.02%, approaching the 75.85% achieved with a noiseless reward. Similar robustness was also observed in Llama models (Llama Team, 2024). These results suggest that even model responses with incorrect final answers may still deserve reward. Because their answers are wrong, *the value worthy of reward likely lies in the reasoning processes.*

---

[1] Our code and scripts are available at `https://anonymous.4open.science/r/NoisyRewards.`

We hypothesize that learning effective reasoning patterns during rollouts is one of the keys to RL improvements. To test this, we introduced **Reasoning Pattern Reward (RPR)**, which rewards the use of certain reasoning phrases (e.g., "first, I need to") regardless of final-answer correctness. Using only RPR—without any correctness supervision, Qwen-2.5-7B achieved 70.21% on MATH-500, comparable to full-verification training. This robustness is consistent across our experiments with Llama, further supporting the validity of our hypothesis. Since no correctness labels were provided and little new knowledge was acquired *in these experiments*, the improvements strongly suggest that the performance gains are due to the reinforcement of useful reasoning patterns.

Section 4 and 5 extend these insights to non-verifiable tasks, focusing on AI assistance generation across diverse user queries. To simulate real-world noise conditions, we trained multiple reward models with varying accuracy levels. We demonstrate that while Qwen and Llama models in non-verifiable tasks exhibit a degree of robustness, it is lower than their robustness in well-pretrained verifiable tasks. Nonetheless, we confirm that reasoning patterns remain influential and can effectively overcome such noise. We employ RPR to calibrate reward models by compensating for potential false negatives. This simple yet effective method introduces minimal overhead and improves the net win rate by up to 30% compared to LLMs trained with vanilla reward models. Furthermore, the RPR-calibrated reward model enables even 3B-parameter small models to achieve notable success in complex tasks where training with vanilla reward models would otherwise collapse.

In summary, our contribution are two-fold:

(1) Our preliminary experiments on verifiable tasks are the first to demonstrate LLMs' robustness to reward noise during post-training and further reveal a key insight: reinforcing effective reasoning patterns is one of primary contributors to RL improvements.

(2) Building on this finding, we focus on open-domain non-verifiable tasks and propose a simple yet effective method that calibrates noisy reward models by explicitly rewarding high-quality reasoning patterns. This approach not only enhances the performance of large language models but also lowers the threshold for enabling effective reasoning in smaller models.

We hope this work offers valuable insights to future research in reinforcement learning for non-verifiable, open-domain tasks.

## 2 RELATED WORKS

**The robustness to reward noise.** (Shao et al., 2025) is a concurrent preprint that also studies robustness to reward noise. Their study, however, is limited in several key aspects: it demonstrates robustness only on Qwen models without extension to other model families, and it does not provide an explanation for the observed robustness. Beyond merely reporting the robustness, our work first identifies one of its underlying causes—the importance of reasoning patterns. Furthermore, we actively leverage this insight by utilizing reasoning patterns to achieve success in challenging open-domain, non-verifiable NLP tasks.

**Reward model accuracy.** An accurate reward model was considered crucial for successful RL (Frick et al., 2024; Lambert et al., 2025; Liu et al., 2025; Zhou et al., 2025). Even in math tasks where rewards are calculated by verification functions, Yeo et al. (2025) proposed that it is beneficial to refine the reward function with a fine-grained approach for accurately evaluating math answers, considering factors such as output length, correctness, and repetition. However, Chen et al. (2024a) found that more accurate reward models do not necessarily lead to stronger LLMs in downstream tasks. Razin et al. (2025) argues that high reward variance is also important for making the reward model a good teacher. Additionally, Wen et al. (2025) suggested that relying solely on accuracy does not fully capture the impact of reward models on policy optimization.

Studying the accuracy of reward models from a realistic perspective, i.e., accounting for noise, offers new insights. We contend that RMs need not be flawless, though calibration to noisy rewards improves evaluation. We provide the first evidence of LLMs' robustness to significant reward noise.

**The role of RL in post-training LLMs.** This paper aligns with recent studies suggesting that pre-trained models already possess the fundamental reasoning abilities needed for complex tasks. Yeo et al. (2025) found that pre-training data often includes long chain-of-thought patterns, establishing a foundation for reasoning. Similarly, Yue et al. (2025) noted that base models can perform similarly

> A conversation between User and Assistant. The user asks a question, and the Assistant solves it. The assistant first thinks about the reasoning process in the mind and then provides the user with the answer. The reasoning process and answer are enclosed within <think> </think> and <answer> </answer> tags, respectively, i.e., <think> reasoning process here </think> <answer> answer here </answer>. User: You must put your answer inside <answer> </answer> tags, i.e., <answer> answer here </answer>. And your final answer will be extracted automatically by the \boxed{} tag. {question} Assistant: <think>

Figure 1: The prompt used in verifiable task training, where the "question" placeholder will be replaced with a specific question.

to RL-post-trained models after multiple attempts at difficult tasks. Gandhi et al. (2025) showed that Qwen models outperform Llama (Llama Team, 2024) models in downstream tasks post-RL, with Qwen models exhibiting natural reasoning behavior. Prior works (AI et al., 2025) demonstrated that reasoning can emerge during pre-training, with models using a reasoning trigger token like "wait" to activate chain-of-thoughts and arrive at the correct answer.

We provide strong evidence that models can achieve peak performance, comparable to those trained with strict verification, by rewarding key reasoning patterns instead of requiring correctness verification. While RL post-training has seen significant progress, our findings highlight the continued importance of pre-training in building advanced LLMs. From a post-training perspective, this also explains why a small amount of high-quality data (Muennighoff et al., 2025) can enhance reasoning abilities, as the foundational capabilities are already present and need effective triggers.

## 3 INSIGHTS VIA MANUALLY INTRODUCED NOISE IN VERIFIABLE REWARDS

### 3.1 LLMS ARE ROBUST TO NOISY REWARDS

Mathematics and multi-choice QA are among the most widely studied domains for reasoning in large language models (LLMs), owing to their clear rule-based rewards and objective evaluation metrics. To explore the RL performance under noisy rewards, we start by manually injecting noise into the RL rewards of verifiable tasks.

**Random reward flip**   We train the model by randomly flipping the reward with a probability $p$, where a reward of 1 is transformed to 0, and vice versa. This flip is applied on a question-wise basis, meaning that if a reward flip occurs for a given question, the rewards for all rollout outputs corresponding to that question will be flipped. Note that flipping rewards on an output-wise basis does not effectively introduce noise. For instance, when an LLM generates multiple correct outputs, some of which are rewarded correctly while others are not, it results in a sparse reward distribution. Such sparsity can only slow convergence and has minimal impact on the model's final performance.

**Training setup**   Most of the training setups follow the approach in (Hu et al., 2025), which provides a simplified framework designed to help LLMs learn to reason. The experiments are based on VeRL (Sheng et al., 2024) framework by Volcengine. The dataset includes 57K high-quality, source-mixture math problems spanning various difficulty levels. In practice, we observed that the model performance plateaued after training on only a subset of the dataset.

The prompts used in these tasks are shown in Figure 1. Notably, the prompt is directly decoded without applying model-specific chat template to avoid potential bias across different model families. While this approach may affect initial accuracy, it does not impact converged performance. Model outputs are extracted from the box tag, normalized in format (e.g., converting fractions to decimals and translating LaTeX answers to plain text), and compared to the ground truth.

In the absence of manual reward noise, the model is given a reward of 1 if the output matches the ground truth and 0 otherwise. For training, we employ vanilla PPO (Schulman et al., 2017) with GAE (Schulman et al., 2018), using $\lambda = 1$ and $\gamma = 1$, with no KL-regularization. The training batch size is 128, with a maximum response length of 4096. The learning rate for the actor is $10^{-6}$, and for the critic, it is $5 \times 10^{-6}$. To ensure stable training, we apply critic warmups for 20 steps, initially training the critic before training the actor model. We set the rollout number to 4. We use Qwen-2.5-7B (Yang et al., 2024) and Llama-3.1-8B-Instruct (Llama Team, 2024) as base models.

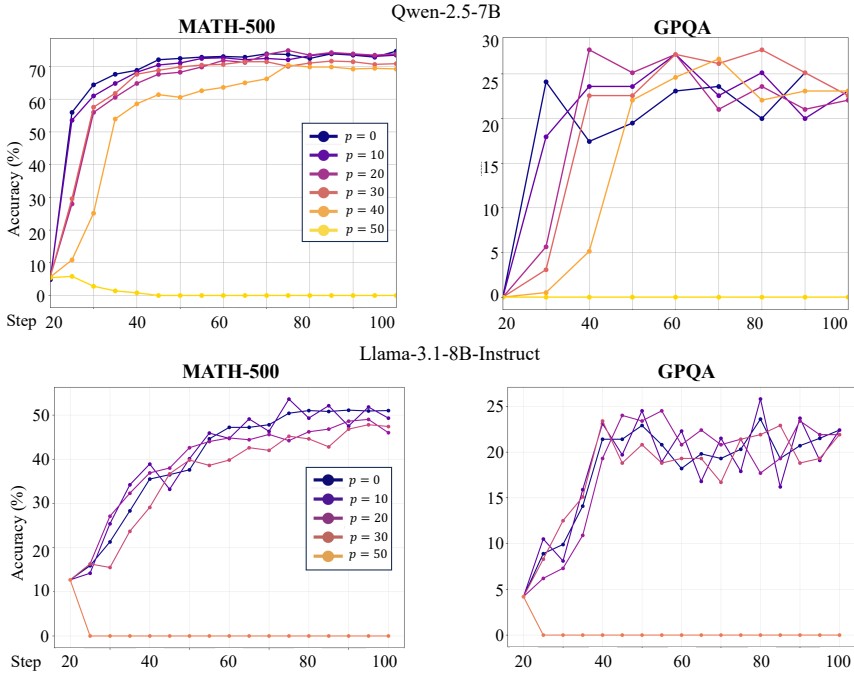

Figure 2: Model performance under varying noise probabilities $p$. Top: Qwen-2.5-7B. Bottom: Llama-3.1-8B-Instruct. Both models show robustness to a large reward noise.

**Evaluation**   To assess the model's reasoning capability on challenging tasks, we utilize two datasets: MATH-500 (Hendrycks et al., 2021), a math dataset, and GPQA (Rein et al., 2023), an out-of-domain dataset relative to the math-centric training data. We report the Pass@1 accuracy dynamics throughout the training.

**Experiments: The impact of** $p$   We train the model with the probability $p$ of noise increasing from 0% to 50%, with intervals of 10%, corresponding to increasingly random reward flips. The results are shown in Figure 2. We only display the first 100 steps, as the performance has already plateaued.

On the MATH-500, the Qwen model exhibits strong robustness, maintaining high performance with a noise level (flip rate) of up to 40%. At this point, its accuracy begins to decline compared to training with no noise, but still yields a peak score of 72.02% versus 75.85%. At lower noise levels, the final performance remains comparable to the noise-free baseline, and convergence occurs at a similar rate. When the flip rate $p$ is increased to 50%, making the reward signal entirely random, training collapses. Similar trends are observed in the GPQA task, although performance fluctuations are more pronounced. The Llama model remains stable up to a noise level of $p = 30 \sim 40\%$. Although still robust, its tolerance to noise is weaker than that of the Qwen models. This is likely related to its inherently weaker baseline capability. Additionally, Figure 13 shows that the small Qwen-2.5-3B model also exhibits strong noise robustness, comparable to the 7B model.

### 3.2 HYPOTHESIS AND VALIDATION: THE ROLE OF REASONING PATTERNS IN PERFORMANCE IMPROVEMENT

Given the surprising result, the key question is why assigning a reward of 1 to outputs with genuinely incorrect answers does not have a significant detrimental effect. Since the answer is incorrect, we hypothesize that the reasoning process itself must be valuable and worth rewarding.

To test this, we conducted an experiment: We first identified $n$ high-frequency phrases that imply certain desired reasoning patterns, such as *"We know that"* and *"First I need to,"* in the outputs of a model trained with $p$=0.[2]   Next, instead of verifying the correctness

---

[2]These phrases also frequently appear in model outputs trained with higher levels of noise.

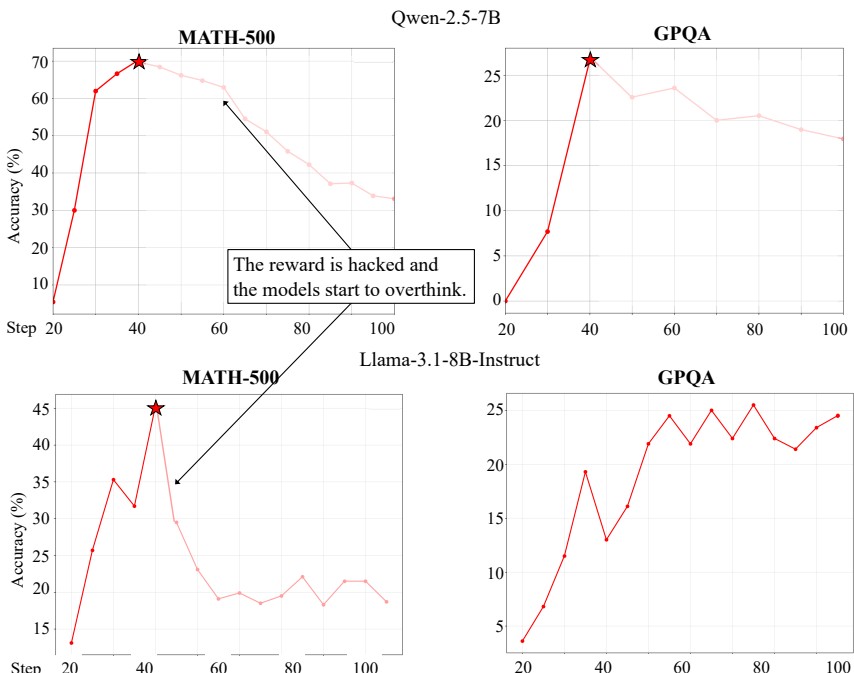

Figure 4: We applied RL post-training using RPR as the sole reward signal. The resulting models achieved high peak performance; however, in some cases, this led to reward hacking, characterized by overthinking and subsequent performance degradation after the peak. As shown in Section 5, combining RPR with other reward signals eliminates reward hacking.

of the answer, the model receives a reward of value $r$ each time a pre-identified reasoning phrase appeared in the output. We name this strategy as ***Reasoning Pattern Reward*** (RPR).

Figure 3 illustrates how RPR works. To prevent the model from hacking the reward by outputting repeated reasoning phrases (e.g., "We know that We know that We know that..."), a repetition penalty (Yeo et al., 2025) is used. Implementation details, including the keyword lists, example code, and hyperparameters $(n, r)$, are provided in Appendix A.

The results are presented in Figure 4. Remarkably, even ***without*** verifying the correctness of reasoning during the training process, the model demonstrates strong reasoning capabilities in the early stages, achieving high performance with only a minimal gap compared to models trained without noise. Therefore, we have demonstrated that:

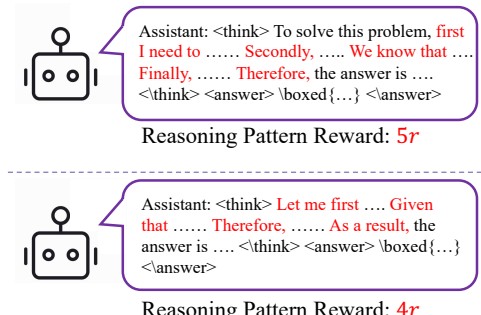

Figure 3: The reasoning pattern reward based on the count of pre-identified high-frequency reasoning phrases.

> Reasoning patterns alone—even without correctness supervision—can elicit high model performance, confirming the importance of learning effective reasoning patterns.

As training progresses, the performance of both models on MATH-500 (and of Qwen on GPQA) declines due to overthinking. Output analysis shows that after several reasoning steps, the model begins to revisit and extend previous thoughts in a cyclical manner. The long reasoning chain finally exceeds the context limit and is truncated before the final answer can be generated. An example is shown in Figure 11, where the model has already arrived at the correct answer but continues to reason, preventing the extraction of the final answer.

These performance drops, caused by reward hacking, are an inherent limitation of using rule-based rewards **alone**, but they do not undermine the importance of reasoning patterns. As we show in §5, this issue is effectively mitigated when RPR is combined with other reward sources.

**Declarations**    To avoid potential misunderstandings, we clarify the following:

(1) **Our focus on reasoning patterns is not a denial of the role new knowledge plays in improving RL performance**, but an investigation of that role is not this paper's focus.

(2) The primary goal of this experiment is to validate the hypothesis that reasoning patterns emerging even in incorrect outputs are valuable, and that reinforcing them improves reasoning ability—not to achieve state-of-the-art accuracy. Therefore, fine-tuning RPR hyperparameters, optimizing keyword lists, or preventing reward hacking are not primary concerns in this section. We also note that the keyword list we collected is broadly applicable across tasks, as shown in Section 5.

# 4    LEARNING TO REASON USING REWARD MODELS OF VARYING ACCURACY

Building on the insights above, now we turn to the open NLP tasks requiring reward models (RMs). Different from manually flipping rewards in verifiable tasks, the noise level in open NLP tasks can be approximately reflected in the varying evaluation accuracies of RMs. We first introduce the data, RM training details, followed by experiments with noisy RL rewards and corresponding findings.

**Dataset**    We use the NVIDIA HelpSteer3 (Wang et al., 2025) dataset, which contains 40.5K multi-domain open-ended questions that require helpful assistance. Each question is paired with two responses, evaluated by multiple annotators for helpfulness, categorized into seven fine-grained levels. There is also a chat history preceding the current question, providing context for the question. The dataset is split into a training set of 38.5K samples and a validation set of 2K samples.

**RM training**    Our RMs are built on a Qwen-2.5-7B model with an added prediction head. We simplify the original seven-level helpfulness scale into a binary classification task: the more helpful response in each pair is labeled as 1, and the less helpful one as 0. For each response, we concatenate it with the chat history as the input to the RM. The prediction head produces a scalar output $s$, and we optimize the model using the MSE between $s$ and the corresponding binary label (Liu et al., 2024a; Zhang et al., 2024a; Deng et al., 2024). The model learns to predict the absolute helpfulness, facilitating further RL, instead of using contrastive learning to compare the relative helpfulness of paired responses. The learning rate is $10^{-6}$, and the RM is trained for 25,000 steps.

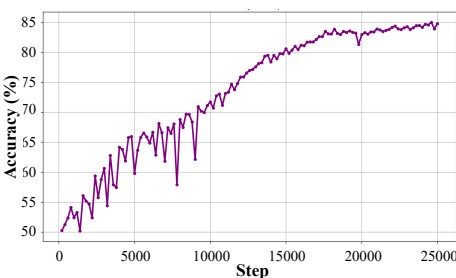

Figure 5: Reward model's accuracy across the training. Checkpoints at specific steps are used for RL experiments.

The evaluation accuracy dynamics of RMs are shown in Figure 5. The best-performing model achieved an evaluation accuracy of 85%. Different RMs with varying accuracies are used in subsequent experiments to simulate the scenarios with different levels of reward noises.

**LLM Training**    The hyperparameters used in this section basically follow those employed in previous verifiable experiments. Training is conducted with Qwen-2.5-7B and Llama-3.1-8B-Instruct on the HelpSteer3 dataset, lasting a total of 200 steps. Figure 6 shows the prompt template, which instructs the model to first carefully consider how to provide useful assistance. It then asks the model to summarize its reasoning and present the final response within the <answer> tag. Importantly, the RM only evaluates the text within the <answer> tag, not the entire output. This approach ensures that the reward pipeline aligns with the one used in previous verifiable experiments.

**Evaluation Settings**    Evaluating open-ended tasks during training presents a much greater challenge than evaluating verifiable problems, due to the lack of objective criteria and the absence of reliable, efficient evaluators. Because our most accurate reward model (RM) is used during training, it cannot be employed for evaluation at test time, as LLMs may learn to hack its preferences. It is also prohibitive to ask more advanced LLMs like ChatGPT or human evaluators to frequently eval-

{Chat history} User: I present a question, and you, the assistant, first thinks about the reasoning process in the mind and then provides the user with the answer. Enclose the reasoning within <think>...</think> tags and the final answer — which should also summarize the reasoning — within <answer>...</answer> tags. For example: <think>Reasoning process here</think> <answer>Answer with summary of reasoning here </answer>. Now, here is my question: {question} Assistant: <think>

Figure 6: The prompt used in the HelpSteer3 task, where the "question" and "chat history" place-holders are filled accordingly.

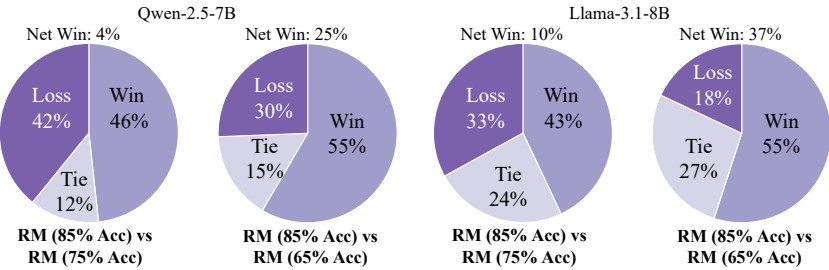

Figure 7: LLMs trained with an 85%-accurate RM performs similarly to using a 75%-accurate RM, but significantly better than using a 65%-accurate RM. The "Net Win" refers to the performance advantage of the former RM over the latter.

uate models during training. As a result, we perform evaluation only after training, using a subset of 200 samples from the evaluation set, assessed by both GPT-4o and human evaluators. Specifically, we compare two models by having GPT-4o and human evaluators assess their responses to the same question. Only text in <answer> tags is used for evaluation.

The prompt used for GPT's evaluation is shown in Figure 15. The evaluation considers factors including helpfulness, informativeness, reasoning, and coverage of user needs. To avoid bias from positional preferences (Liu et al., 2024b; Chen et al., 2024b; Zhang et al., 2024b) in language models, ChatGPT-4o evaluates an output pair twice for the same question, each time with a different order. A model's response may result in a win, loss, or tie relative to the other model's response, with the results presented in pie charts. In the main text, we report GPT evaluation scores, as they are more reproducible for the community. Details on human evaluation—guidelines, results, and inter-evaluator agreement measured by Fleiss' Kappa (Fleiss et al., 1971)—are provided in Appendix B. There, we show that human evaluation aligns with GPT assessments, with evaluators demonstrating moderate to substantial agreement.

**Experiment: Comparing LLMs trained with RMs of various accuracies** We compare the performance of Qwen and Llama trained with reward models (RMs) of varying accuracies: 85%, 75%, and 65%. The results are presented in Figure 7. **Two models perform comparably when trained with reward models that are 75% and 85% accurate, indicating a degree of robustness to reward noise. However, their robustness is less pronounced than what has been observed in verifiable tasks**, since the models trained using the 65%-accurate RM shows a significant decline in downstream performance.

This is because, beyond accuracy, the magnitude and distribution of non-verifiable reward scores are also critical. In contrast to verifiable problems—where rewards are typically binary (e.g., 0 or 1)—RMs, especially less accurate ones, tend to produce scores clustered around 0.5, even when they make correct classifications. This clustering reflects underlying model uncertainty. This effect is exhibited in the reduced variance of reward outputs from lower-accuracy RMs: on a validation set, the score variances are 0.1937, 0.1161, and 0.0672 for the 85%, 75%, and 65%-accurate RMs, respectively. A more accurate RM pushes scores further from the decision boundary, helping to avoid both over- and under-estimated rewards. These observations align with findings by (Razin et al., 2025), who emphasized that higher variance is also a key factor in RM effectiveness. In summary, both the lower accuracy and lower variance of the 65%-accurate RM likely contribute to its weak downstream performance.

RMs with greater than 80% accuracy are often attainable in practice, and our experiments offer relief for real-world applications, where concerns about RM accuracy are prevalent. However, as

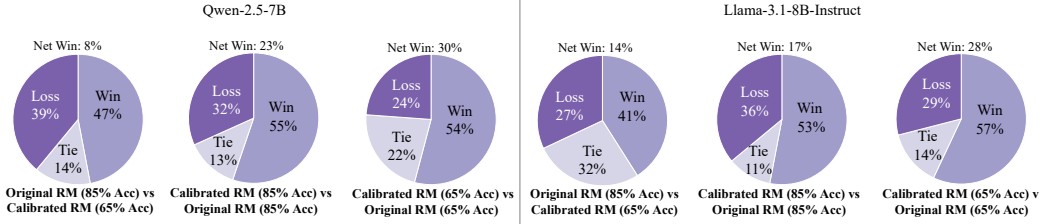

Figure 8: Reward noise calibration effectively enhances downstream performance.

demonstrated by our RM with 65% accuracy, some noisy RMs are indeed inadequate for practical use as the sole source of reward. Recognizing the importance of reasoning patterns, we propose a method for calibrating RMs with RPR (Section 3). This approach overcomes the performance ceiling imposed by the limitations of the reward models at hand.

## 5 CALIBRATING NOISY RMS WITH REASONING PATTERN REWARD

We investigate whether the Reasoning Pattern Reward (RPR; Section 3) can help calibrate weak RMs to obtain better LLM performance. There are two potential ways to calibrating noisy rewards based on the value of reasoning patterns:

**1. Compensatory reward for underestimated responses.** When an RM produces false negatives, assigning a low score to an "objectively" good response, we give it some compensation. [3] We assume that responses that display better reasoning patterns are likely to be closer to the "objectively" good ones. Therefore, we reuse the RPR as the compensation reward.

**2. Discounting for overestimated responses.** Conversely, when an RM provides false positive results, that is, it incorrectly assigns a high score to a "objectively" poor response, we can apply a discount to RM scores. However, this situation is more complex than discounting false negatives. The main challenge is determining the appropriate discount factor. For instance, if a response receives a full score but lacks key reasoning phrases, should its reward be near zero? Setting it too low could overemphasize reasoning pattern rewards, leading to overthinking and performance collapse, as discussed in Section 3. This remains an open research question: how can we effectively calibrate an RM when the noisy reward is a false positive?

Consequently, we only introduce the first method to calibrate the RM model: When the RM outputs a score lower than a pre-defined $\tau$, we calculate an RPR score only for the thought text (enclosed in <think> tags), while text in <answer> tags is not considered. This RPR score is added to the RM output, scaled by a weight $\alpha$. This calibration incurs no additional time or memory costs.

Note that in our approach, RPR compensates not only for false negatives but also for true negatives. This is not problematic, as we demonstrated in Section 3 that true negative responses still contain valuable reasoning patterns and are therefore worth rewarding. Another potential concern is that using RPR as the sole reward signal might lead to performance collapse (Figure 2). However, when RPR is used as an auxiliary signal rather than the sole reward, LLMs are trained effectively without such collapse in the following experiments.

**Experiment: RPR-calibrated RMs enhance RL post-training**  We use RMs with accuracies of 65% and 85% to conduct several comparisons: (1) Calibrating the RMs using RPR, applying it to post-training LLMs, and comparing their performance with models trained solely with the original RMs. (2) Comparing an LLM trained with a 65%-accurate RM calibrated by RPR to a model trained with an 85%-accurate RM. We set the threshold $\tau$ to 0.5 and $\alpha$ to 0.1. The choice of $\alpha$ is discussed in Appendix D. The results in Figure 8 demonstrate the effectiveness of RPR calibration:

1. Qwen trained with the calibrated 65%-accurate reward model lags behind the 85%-accurate counterpart by a mere 8%—an improvement from an initial 25% gap, highlighting the substantial gains achieved through calibration. The Llama model exhibits a similar trend.

---

[3]By "objectively," we refer to whether rewarding the response eventually improves performance on the test set. If it does, the response should be considered good, at least from a deep learning perspective.

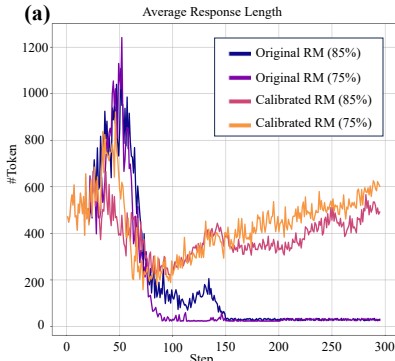 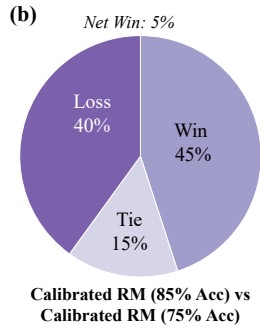

Figure 9: (a) Average response length of Qwen-2.5-3B during training with original vs calibrated RMs. The calibrated RMs successfully enable this small-scale model to perform reasoning, whereas the original RMs fail. (b) Experiment 3 using Qwen-2.5-3B models trained with calibrated RMs. There is no pie chart data for original RMs because they failed to train a viable model for comparison.

2. Calibrating noisy RMs boosts downstream LLM performance, outperforming LLMs trained with original RMs. Even the 85%-accurate RM continues to improve after RPR calibration. RPR calibration addresses the limitations of RMs at hand.

Notably, the improvements observed are not due to an increase in reward score variance (0.1889 and 0.0653 for the 85% and 65%-accurate RMs post-calibration), as variance actually decreases slightly. We provide output examples from models trained with both the original and calibrated RMs in Appendix C.

**Experiment: RPR-calibrated RMs enable small models to effectively reason** In the Help-Steer3 task, vanilla RL fails to enable Qwen-2.5-3B to perform effective reasoning. We observe that response lengths initially increase but then rapidly collapse to just a few tokens. This pattern of "first-reason-then-collapse" has also been observed in (Pan et al., 2025), where an LLM is trained to reason on tasks beyond its initial capabilities. However, the underlying mechanisms driving this length dynamics remain understudied. In contrast, when trained with RPR-calibrated RMs (accuracy > 75%), Qwen-2.5-3B exhibits clear reasoning behaviors. As shown in Figure 9(a), the response lengths differ significantly between models trained with original versus calibrated RMs. These results echo the finding that calibrated RMs more effectively evoke reasoning abilities in large language models. Subfigure (b) shows that using an 85%-accurate RM yields only a 5-point improvement in net win rate over the 75%-accurate RM—mirroring observations in 7B models' robustness to RM noise.

Figure 12 and Figure 14 present two sample outputs from Qwen-2.5-3B trained with the 85%-accurate RM, demonstrating that we have successfully elicited the basic reasoning capabilities of this small-scale model, despite some imperfections in these outputs. In Figure 12, the Chinese query asks for the creation of a PowerPoint file to teach primary school students about statistical charts. The 3B model engages in step-by-step reasoning to generate a coherent PowerPoint structure and follows through on its plan. In Figure 14, the model processes a complex chat history and solves a physics problem, despite not being explicitly trained for mathematics or physics.

## 6 CONCLUSIONS

Enhanced reasoning ability through reinforcement learning in non-verifiable domains remains relatively understudied compared to the well-explored verifiable tasks. We make three key contributions: we first report LLMs' robustness to reward noise during RL post-training; second, through experiments with reasoning pattern rewards (RPR) across both verifiable and non-verifiable NLP tasks, we validate that reasoning patterns play a critical role in RL-driven improvements; third, focusing on real-world non-verifiable tasks, we show that using RPR to calibrate noisy reward models reduces false negative rewards—this not only pushes LLMs to higher performance levels, overcoming limitations of existing reward models, but also enables small models to effective reason. Our contributions provide insights for improving post-training techniques.

## STATEMENTS ON ETHICS, REPRODUCIBILITY, AND LLM USAGE

This paper does not raise special ethical issues. For reproducibility, we used public data and RL framework. We commit to open-sourcing the full codebase. We used LLMs solely for typo checking.

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

# A  THE DESIGN OF REASONING PATTERN REWARDS

Similar to human reasoning where coherent thought relies on a general set of common transitional phrases and logical connectors rather than an extensive list of keywords, our design for the keyword list and its size $n$ reflects this principle.

In all experiments, we set $n = 40$ for RPR. This value was not specifically tuned; empirical observations indicated that $n \geq 40$ leads to a convergence rate similar to that achieved under full correctness verification. Moreover, $n = 40$ already captures a substantial portion of generally effective reasoning patterns. For simplicity, each matched pattern is assigned a reward value of $r = \frac{1}{n} = 0.025$.

An example implementation of RPR is provided in Figure 10. To mitigate the risk of reward hacking and overthinking, we allow a maximum of 10 keyword matches per instance.

The RPR keyword lists, which can also be found in Figure 10, were collected from outputs of a Qwen-2.5-7B model trained on mathematical datasets during rollouts. The reasoning patterns exhibited in these keywords were verified to generalize across both tasks (from mathematical domains to non-verifiable NLP tasks) and model architectures (e.g., Llama). Experimental results validating these claims are presented in §3, §4, and §5.

A potential concern is the risk of reward hacking with RPR. However, experiments in §5 demonstrate that this occurs only when RPR is used as the sole reward signal. In practice, when combined with a reward model, RPR provides positive benefits. Since our core focus is on improving scenarios where reward models have limitations—and *since **using RPR alone** serves only for validation purposes in §3 and is neither a proposed method nor recommended in practice*—the risk of hacking is effectively mitigated in real-world applications. Enhancing the robustness of rule-based rewards like RPR to eliminate potential hacking risks remains an important direction for future work.

# B  HUMAN EVALUATION

## B.1  GUIDELINES

We recruited three graduate students with expertise in model evaluation. Each evaluator spent approximately 8 hours completing all tasks and earned $70 USD. The human evaluation was granted by our institute, with the payment slightly above the standard wage for graduate students working in AI companies in our country. Below is the guideline for human evaluators:

**Guideline for Evaluating Responses:**

Your task is to determine which of the two responses better addresses the user's latest request.

**Steps to Follow:**

- **Review the Conversation History:** Carefully read the conversation history provided. The user's most recent question will be the last message, and that is the request you need to evaluate the responses against.
- **Examine the Two Responses:** You will be presented with two possible replies from two AI assistants (Response #1 and Response #2).
- **Criteria for Evaluation:** Evaluate each response based on the following factors:
  - **Helpfulness:** Does the response directly answer the user's request? Is it practical and useful?
  - **Amount of Information:** Does the response provide sufficient details to address the request thoroughly?
  - **Clarity and Coherence:** Is the response easy to understand, and does it present information logically?
  - **Thoroughness:** Does the response cover all aspects of the user's request? Is anything missing or incomplete?
- **Avoid Quick Judgment:** We will randomize the response order from two models. You cannot infer which one is always better based on the order. Also, don't assume one response is better simply because it's shorter or longer.

```
1   from collections import Counter
2   def calculate_ngram_repetition_penalty(text, n):
3       words = text.split()
4       ngrams = [tuple(words[i:i+n]) for i in range(len(words) - n + 1)]
5       ngram_counts = Counter(ngrams)
6       total_ngrams = len(ngrams)
7       repeated_ngrams = sum(1 for count in ngram_counts.values() if count
            > 1)
8       repetition_penalty = repeated_ngrams / total_ngrams if total_ngrams
            > 0 else 0
9       return repetition_penalty
10
11  def reasoning_pattern_reward(solution):
12      reason_pos = solution.find("Assistant: <think>")
13      solution_str = solution[think_pos + len("Assistant: <think> "):]
14      score = 0
15      solution_str = solution.lower()
16      score += float("i need to" in solution_str)
17      score += float("we need to" in solution_str)
18      score += float("wait" in solution_str)
19      score += float("alternatively" in solution_str)
20      score += float("let me check" in solution_str)
21      score += float("let me see" in solution_str)
22      score += float("let's focus on" in solution_str)
23      score += float("we know that" in solution_str)
24      score += float("we can observe " in solution_str)
25      score += float("we can see " in solution_str)
26      score += float("let me try" in solution_str)
27      score += float("let's try" in solution_str)
28      score += float("let us try" in solution_str)
29      score += float("first," in solution_str)
30      score += float("firstly," in solution_str)
31      score += float("next," in solution_str)
32      score += float("finally," in solution_str)
33      score += float("let us first" in solution_str)
34      score += float("let's first" in solution_str)
35      score += float("let me first" in solution_str)
36      score += float("try again" in solution_str)
37      score += float("still not" in solution_str)
38      score += float("not working" in solution_str)
39      score += float("not correct" in solution_str)
40      score += float("does not work" in solution_str)
41      score += float("doesn't work" in solution_str)
42      score += float("makes sence" in solution_str)
43      score += float("since we" in solution_str)
44      score += float("because we" in solution_str)
45      score += float("consequently" in solution_str)
46      score += float("as a result" in solution_str)
47      score += float("thus" in solution_str)
48      score += float("therefore" in solution_str)
49      score += float("hence" in solution_str)
50      score += float("so that" in solution_str)
51      score += float("thereby" in solution_str)
52      score += float("if we" in solution_str)
53      score += float("given there" in solution_str)
54      score += float("for instance" in solution_str)
55      score += float("for example" in solution_str)
56      score = min(10, score)
57      score /= 40
58      score -= calculate_ngram_repetition_penalty(solution_str, 20)
59      return score
```

Figure 10: An example code of reasoning pattern reward without checking answer correctness.

A conversation between User and Assistant. The user asks a question, and the Assistant solves it. The assistant first thinks about the reasoning process in the mind and then provides the user with the answer. The reasoning process and answer are enclosed within <think> </think> and <answer> </answer> tags, respectively, i.e., <think> reasoning process here </think> <answer> answer here </answer>. User: You must put your answer choice (a single capital letter) inside <answer> </answer> tags, i.e., <answer> answer here </answer>. And your final answer will be extracted automatically by the \boxed{} tag. Dienes are organic compounds with two adjacent double bonds in their structure, and they exhibit unique reactivity due to their conjugated pi-electron system. They play a significant role in organic chemistry and are involved in various chemical reactions and natural processes.

Among the given options which one is the possible reactant (A) for the given reaction also mention the correct sequence of the dienes according to their reactivity ( most reactive to least reactive) B.

Cyclohexene + A ---> 8,8-diiodobicyclo[4.2.0]octan-7-one

(B) 1. 2,3-dimethylbuta-1,3-diene, 2. (2E,4E)-hexa-2,4-diene, 3. cyclopenta-1,3-diene, 4. (2Z,4Z)-hexa-2,4-diene

A. A = 2,2-diiodoethen-1-one, B = 3, 1, 2, 4
B. A = 4,4-diiodocyclobut-2-en-1-one, B = 3, 1, 2, 4
C. A = 2,2-diiodoethen-1-one, B = 4, 2, 1, 3
D. A = 4,4-diiodocyclobut-2-en-1-one, B = 4, 2, 1, 3

Assistant: <think> First, I need to understand what we know about dienes and how they react with other compounds. We know that dienes have two adjacent double bonds, so let's see if we can figure out which of these options contains a diene. Additionally, we need to determine the correct sequence of dienes based on their reactivity, so let me try to recall some information about.

Firstly, since we are looking at a reaction involving cyclohexene, it seems like we might be dealing with an addition reaction where one of the dienes would add across the double bond of cyclohexene. Next, I need to consider the structure of each option provided and identify any dienes present within them. Furthermore, since we want to find the most reactive diene first, I should look into the properties of different types of dienes and their reactivity patterns.

Next, let's try to analyze the structures of these compounds and see if we can spot any dienes among them. Also, we need to figure out how these dienes would react with cyclohexene and form the product mentioned in the question. Lastly, we need to determine the correct order of reactivity for these dienes.

Firstly, let us examine option A because it has a simple structure and contains a diene. Now, let us check if this diene could react with cyclohexene to produce the desired product. Secondly, let us compare the structures of the other options to see if they contain dienes as well. Finally, we need to establish the correct sequence of reactivity for these dienes.

Firstly, let us focus on option A, which includes 2,2-diiodoethen-1-one and 3,1,2,4 as its components. Since we know that dienes are characterized by having two adjacent double bonds, let us try to identify those in this option. Upon closer inspection, we notice that there is indeed a diene present in this option, specifically 2,2-diiodoethen-1-one. This compound contains two adjacent double bonds, making it a diene. Therefore, we can conclude that option A contains a diene. However, we still need to determine if this diene can react with cyclohexene to form the required product.

Next, let us examine option B, which consists of 4,4-diiodocyclobut-2-en-1-one and 3,1,2,4 as its components. Again, we need to check if this option contains a diene. After analyzing the structure, we find that there is no diene present in this option. Hence, we can eliminate option B as a potential reactant.

Now, let us move on to option C, which comprises 2,2-diiodoethen-1-one and 4,2,1,3 as its components. Similar to our previous analysis, we observe that there is a diene present in this option, namely 2,2-diiodoethen-1-one. Thus, we can confirm that option C also contains a diene. However, we must now determine if this diene can react with cyclohexene to yield the desired product.

Lastly, let us analyze option D, which includes 4,4-diiodocyclobut-2-en-1-one and 4,2,1,3 as its components. Once more, we need to verify if this option contains a diene. Upon examination, we realize that there is no diene present in this option. Consequently, we can rule out option D as a viable reactant.

Having analyzed all the options, we have identified that both options A and C contain a diene. Therefore, we need to decide which one is the correct reactant for the given reaction. To do this, we need to consider the reactivity of these dienes and determine which one is more likely to participate in the reaction.

Firstly, let us think about the general reactivity pattern of dienes. Generally speaking, dienes with more substituted double bonds tend to be more reactive than those with fewer substituents. Moreover, dienes with electron-donating groups attached to the double bond are typically more reactive than those without such groups. With this knowledge in mind, let us compare the structures of the dienes present in options A and C.

Upon comparing the structures, we notice that the diene in option A has two iodine atoms attached to the double bond, whereas the diene in option C does not have any substituents on the double bond. Consequently, we can infer that the diene in option A is likely to be more reactive than the one in option C due to the presence of the electron-withdrawing iodine atoms.

Therefore, we can conclude that option A is the most probable reactant for the given reaction. Now, we need to establish the correct sequence of reactivity for these dienes.

Secondly, let us think about the reactivity of dienes in general. Typically, dienes with more substituted double bonds tend to be more reactive than those with fewer substituents. Furthermore, dienes with electron-donating groups attached to the double bond are usually more reactive than those without such groups. Based on this information, we can deduce that the diene in option A, which has two iodine atoms attached to the double bond, is likely to be the most reactive diene among the ones we have considered. Next, we need to determine the reactivity of the diene in option C.

Finally, let us consider the reactivity of the diene in option C. As previously mentioned, the diene in option C lacks any substituents on the double bond. Consequently, we can infer that this diene is likely to be less reactive than the one in option A. Thus, we can conclude that the diene in option C is the least reactive diene among the ones we have analyzed.

Hence, we can establish the correct sequence of reactivity for these dienes as follows: the diene in option A is the most reactive, followed by the diene in option C, which is the least reactive.

Lastly, we need to determine the correct sequence of the dienes according to their reactivity. Since we have already established that the diene in option A is the most reactive and the diene in option C is the least reactive, we can conclude that the correct sequence of reactivity for these dienes is option A, then option C.

Thus, we need to select the correct option from the given choices. Given that we have determined that option A is the correct reactant and the correct sequence of reactivity is option A, we can now proceed to choose the appropriate answer.

Hence, we need to select the option that matches our findings. Since we have concluded that option A is the correct reactant and the correct sequence of reactivity is option A, we can deduce that the correct answer must be option A.

Therefore, we need to choose the option that corresponds to our findings. Hence, we need to select the option that matches our conclusion. Consequently, we can conclude that the correct answer is option A.

Finally, we need to express our final answer using the provided format. Thus, we need to enclose our answer within the specified tags.

Hence, I shall now provide my final answer in the required format.

Thus, we can conclude that the correct answer is \boxed{\text{A}}.

Therefore, we can write our final answer as follows: $\therefore \text{The correct answer is } \boxed{\text{A}}$. Hence, we can express our final answer as $\boxed{\text{A}}$.

Figure 11: An example of output in the later stage of RL math training, where only the reasoning pattern reward is used without correctness verification. The model has arrived at the correct answer "A," but due to the ongoing reasoning process, the <think> tag remains open, causing the output length to reach the limit and preventing the correct answer from being generated in answer tags.

**Question:**
幫我設計教學用ppt，要帶小學生學習[統計圖表]，整體PPT結構要有導入活動、開展活動、總結活動，這三個活動中每個活動要有引發學生探究的問題並附上一張圖片來說明問題情境、概念講解(可以詳細點)、以及數學多重表徵實體操作活動(詳列步驟)、最後面要有形成性評量檢核點，每個檢核點，前面置放"☐"符號置左，檢核點要有知識、技能、態度面向。整個ppt請用markdown標記語言格式，三級結構，分數部分請以a/b形式比較看得懂。\n 圖片部分，請先將 QUERY，翻譯成英文，再由這個網站全部英文輸入取得 (https://source.unsplash.com/960×640/?{ QUERY})。使用Markdown 語法來顯示圖片

**Answer:**
First, I need to plan the structure of the PPT for teaching statistics to primary school students. This includes creating a clear introduction, engaging activities, and a conclusion. Each section should include a problem to stimulate inquiry, concept explanation, hands-on activities, and assessment points. For images, I will use a combination of relevant images and custom ones generated using the provided website. Finally, I will format the PPT using Markdown with appropriate headers and lists.

Reasoning process:
1. Begin by drafting an introduction that captures the students' attention and explains the importance of learning about statistical graphs.
2. Design the "engage" section with a problem that encourages students to explore different types of graphs, such as bar charts or pie charts, and how they represent data.
3. Develop the "explain" section by providing clear definitions and examples of each type of graph, using visual aids to enhance understanding.
4. Create the "do" section with hands-on activities where students can create their own graphs using real-life data sets, reinforcing the concepts learned.
5. Prepare the "summarize" section with a review of the key points and encourage students to reflect on what they've learned.
6. Incorporate assessment points at the end of each activity, detailing what students should be able to demonstrate in terms of knowledge, skills, and attitudes.
7. Generate images for the PPT by translating search queries into English and using the website to fetch appropriate images.

Final Answer:
```markdown
# Teaching Statistics to Primary School Students

## Introduction
Welcome to our lesson on statistical graphs! Understanding how to interpret and create these visual representations is crucial for analyzing data in everyday life.

## Engage
**Question:** How can we visually compare the favorite fruits of our classmates?

**Image:** [Image of students discussing their favorite fruits]

## Explain
**Concept:** What are bar charts and pie charts? How do they differ?

**Image:** [Bar Chart Image]
[Pie Chart Image]

**Definition:** A bar chart uses bars to show comparisons between different categories of data. A pie chart shows the proportion of each category in relation to the whole.

## Do
**Activity:** Create a bar chart and a pie chart based on the following data:
- 15 students like apples
- 10 students like bananas
- 5 students like oranges

**Image:** [Example Bar Chart]
[Example Pie Chart]

## Summarize
**Review:** What did you learn about bar charts and pie charts? Can you describe their differences?

## Assessment Points
- **Knowledge:** Can you identify which type of graph is best suited for comparing quantities?
- **Skills:** Can you create a bar chart or pie chart from given data?
- **Attitudes:** Are you curious about how statistics can help us understand the world better?

☐ Knowledge ☐ Skills ☐ Attitudes
```

This PPT structure ensures a comprehensive learning experience for the students, combining theoretical knowledge with practical application, all while keeping the content engaging and interactive.

Figure 12: Our calibrated RMs successfully elicit Qwen-2.5-3B's reasoning ability, whereas the original RM fails to do so. This figure presents 1 of 2 output cases. The Chinese question translates to: "Please design a teaching PowerPoint for teaching elementary school students about [statistical charts]. The overall structure of the PPT should include three parts: introductory activity, developmental activity, and summary activity. Each of these activities must include: A question that triggers student inquiry, along with an image illustrating the context of the problem. A concept explanation (detailed). A math hands-on activity using multiple representations, with a clearly listed step-by-step process. At the end of the PPT, include a formative assessment checklist. Each checklist item should start with the symbol "☐" aligned to the left and should address three aspects: knowledge, skills, and attitudes. Please present the entire PPT in Markdown format with a three-level heading structure. For fractions, use the a/b format as it is easier for me to understand. As for the images, first translate the QUERY into English, and then use the following website with an English query to retrieve the images:https://source.unsplash.com/960x640/?QUERY Use Markdown syntax to display the images."

After evaluating both responses, decide which one is more helpful overall. You can choose #1 is better, #2 is better, or they tie with each other. Write the evaluation, as well as reasons.

## B.2 RESULTS AND INTER-ANNOTATOR AGREEMENT

In Figure 16, we present the averaged human evaluation results for Experiments 3 and 4 in Section 4. Each figure also reports inter-evaluator agreement $\kappa$, with all experiments demonstrating

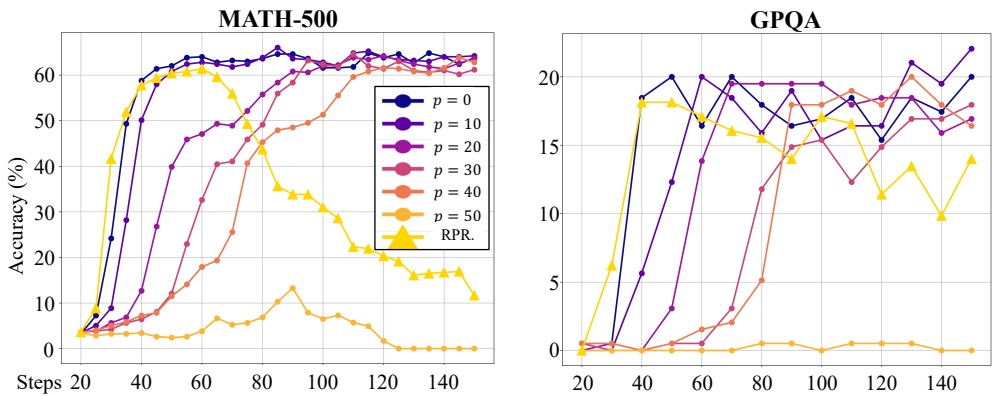

Figure 13: Qwen-2.5-3B: Impact of $p$ and RPR-only training.

**Chat History:**
User: What was the total length of the Circus Maximus?\nHow many laps were in a traditional chariot race?\nHow long would a chariot race take on average?

Assistant: The Circus Maximus in ancient Rome was approximately 621 meters (2,037 feet) in length. It was a massive stadium used for chariot racing and other public events.\n\nIn a traditional chariot race, charioteers would have to complete 7 laps around the Circus Maximus. The starting point was at the closed end of the track, and the chariots would race towards the open end, then turn around a metae (turning post) and head back in the other direction.\n\nThe average time for a chariot race in the Circus Maximus would depend on various factors such as the number of competitors, the condition of the track, and the skill of the charioteers. However, a typical race could last anywhere from 8 to 15 minutes

**Question:**
If a car is driving on a 45° bank 1 mile oval track, how fast does the car have to be to avoid falling off the track?

**Answer:**
To ensure the car stays on the banked track, we need to balance the forces acting on it. The forces include gravity, the normal force, and friction. On a banked curve, the normal force acts at an angle, and we can resolve the gravitational force into components parallel and perpendicular to the surface. The component parallel to the surface provides the necessary centripetal force for circular motion. The formula for centripetal force is $F_c = \frac{mv^2}{r}$, where $m$ is the mass of the car, $v$ is the velocity, and $r$ is the radius of the curve. The radius $r$ can be calculated from the length of the track and the angle of the bank using $r = \frac{L}{2\sin(\theta)}$, where $L$ is the length of the track and $\theta$ is the angle of the bank. Substituting $r$ into the centripetal force formula and equating it to the component of the gravitational force parallel to the surface ($mg\sin(\theta)$), we can solve for the minimum velocity $v$. Given that the length of the track $L = 1$ mile and the angle of the bank $\theta = 45^\circ$, the radius $r = \frac{1}{2\sin(45^\circ)} = \frac{1}{2 \cdot \frac{\sqrt{2}}{2}} = \frac{1}{\sqrt{2}}$. Therefore, the minimum velocity $v$ is given by $v = \sqrt{\frac{2mg\sin(\theta)}{m}} = \sqrt{2g\sin(\theta)}$. Since $\sin(45^\circ) = \frac{\sqrt{2}}{2}$, we have $v = \sqrt{2g \cdot \frac{\sqrt{2}}{2}} = \sqrt{g\sqrt{2}}$. Using $g \approx 32 \, \text{ft/s}^2$, we find $v \approx \sqrt{32 \cdot \sqrt{2}} \approx \sqrt{60.8} \approx 7.8 \, \text{ft/s}$. Converting this to miles per hour (since 1 mile = 5280 feet), we get $v \approx 7.8 \times \frac{3600}{5280} \approx 51.4 \, \text{mph}$. Therefore, the car must be traveling at least 51.4 mph to avoid falling off the track.

Final Answer: The car must be traveling at least 51.4 mph to avoid falling off the track.

Figure 14: This figure presents the second of two output cases from Qwen-2.5-3B, trained with the calibrated 85%-accurate RM. The model demonstrates reasoning to solve a physics problem.

moderate ($0.4 < \kappa \le 0.6$) to substantial ($0.6 < \kappa \le 0.8$) consistency among evaluators. A key distinction between human evaluations and those from GPT-4o is that human judges exhibit stronger discriminatory ability, resulting in fewer comparisons marked as "ties." Nonetheless, the overall conclusions—such as the net win ratios and the impact of calibration—align closely with the GPT-based evaluations. Therefore, we do not repeat Takeaways and conclusions here.

Your task is to help me determine which of two responses is more helpful in addressing the user's latest request.
First, I will provide you with the conversation history between the user and the AI assistant, where the last statement is the user's most recent request.
Then, I will show you the two assistant replies: response #1 and response #2.
Please evaluate which response is better based on factors such as helpfulness, the amount of information provided, the thoroughness of the reasoning, and overall coverage of the user's needs.
Please check two responses carefully, do not casually answer that #1 is better.
**Make sure to clearly state whether '#1 is better' or '#2 is better' or 'tie' AT THE END OF YOUR ANSWER**.

Here is the conversation history:
{chat}

Response #1:
{res1}

Response #2:
{res2}

Figure 15: The evaluation prompt for GPT, designed according to the core guidelines for human annotators. The placeholders will be replaced with user-assistant chat history and two models' responses.

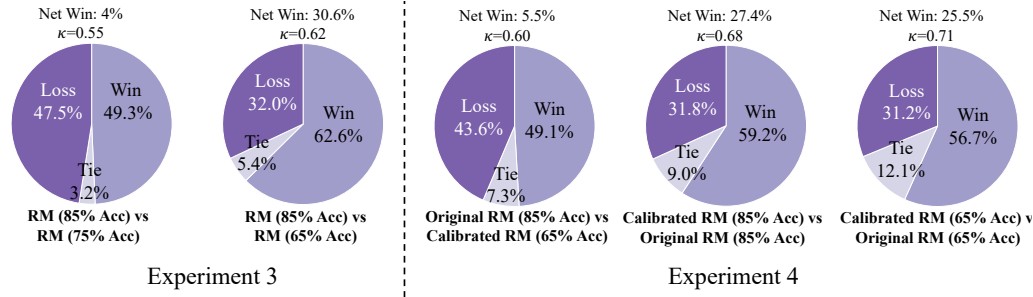

Figure 16: Human evaluation results and agreements.

## C  CASE STUDIES

Figures 18 and 19 show outputs from Qwen-2.5-7B trained with the calibrated and original 85%-accurate RMs, respectively. With RPR, the generated code and comments are more detailed, and both the main function and interaction loop are more comprehensive compared to the single test case produced by models trained without RPR. The reasoning process is also more thoroughly articulated.

Figures 20 and 21 illustrate Qwen-2.5-7B trained with calibrated and original 65%-accurate RMs, respectively. Compared to models trained with 85%-accurate RMs, both outputs here fail to explicitly move the model to the GPU. However, the model trained with the calibrated 65%-accurate RM correctly implements a chatbot using the transformers pipeline API, which implicitly moves the model to the GPU. As a result, the model trained with the original 65%-accurate RM performs slightly worse in terms of helpfulness. It is uncommon for an assistant to build a chatbot using the transformers pipeline—an approach that is both concise and effective—suggesting that Qwen models have acquired substantial knowledge during pretraining.

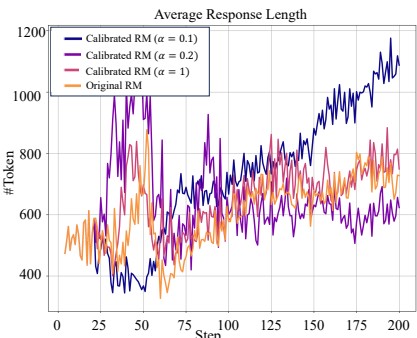

Figure 17: $\alpha = 0.1$ enables Qwen's effective reasoning in HelpSteer3 task. These experiments use the 85%-accurate RM.

## D    RPR WEIGHT IN CALIBRATION

In Section 5, we set $\tau = 0.5$ since the RMs are binary classifiers. For the scaling factor $\alpha$, we initially used a value of 1, but observed little difference in training dynamics—such as average response length—compared to using the original RMs. We then experimented with values of $\alpha$ ranging from 1 to 0.1. Notably, reducing $\alpha$ to 0.1 led to a significant increase in average response length (see Figure 17), suggesting a shift in the model's reasoning behavior. Based on these findings, we set $\alpha = 0.1$.

**Question:** Generate python code for creating a chatbot using a large language model

**Answer:** To create a chatbot using a large language model, we need to follow a structured approach that involves several steps, including choosing the right model, preparing the data, setting up the chatbot framework, and testing the chatbot. Below, I will outline the reasoning process and provide a Python code example that demonstrates how to create a simple chatbot using the Hugging Face Transformers library, which provides access to various large language models. This example will use the GPT-2 model as an example, but you can replace it with other models as needed.

### Reasoning Process

1. **Choose the Right Large Language Model**: Select a suitable pre-trained large language model. For this example, we'll use the GPT-2 model, which is known for its ability to generate human-like text. Other options include GPT-3, BERT, or other models depending on the specific requirements and the tasks the chatbot will perform.

2. **Set Up the Environment**: Ensure you have the necessary libraries installed. This includes the Hugging Face Transformers library for accessing the pre-trained models and tokenizers, and possibly other libraries for additional functionalities like handling user input and output.

3. **Load the Pre-Trained Model and Tokenizer**: Use the Transformers library to load the pre-trained model and its corresponding tokenizer. The tokenizer is crucial for converting text into tokens that the model can understand and for converting the model's outputs back into human-readable text.

4. **Define the Chatbot Logic**: Implement the logic for handling user input, generating responses using the model, and managing the conversation flow. This might involve simple rules-based systems or more complex natural language understanding (NLU) techniques.

5. **Test and Iterate**: Test the chatbot with various inputs to ensure it behaves as expected. Iterate on the model selection, data preparation, and chatbot logic based on the test results and any feedback.

### Summary of Reasoning

The process of creating a chatbot using a large language model involves selecting an appropriate model, setting up the necessary environment, loading the model and tokenizer, defining the chatbot's interaction logic, and thoroughly testing the chatbot to refine its performance. The key is to leverage the capabilities of the large language model for generating coherent and contextually relevant responses while ensuring the chatbot is user-friendly and effective in its intended use case.

### Python Code Example

Here is a Python code example that demonstrates these steps using the Hugging Face Transformers library to create a simple chatbot with the GPT-2 model:

```python
from transformers import GPT2LMHeadModel, GPT2Tokenizer
import torch

# Step 1: Choose the right large language model
model_name = 'gpt2'  # You can change this to other models like 'gpt2-medium', 'gpt2-large', etc.

# Step 2: Set up the environment
# Ensure you have the necessary libraries installed: transformers, torch

# Step 3: Load the pre-trained model and tokenizer
tokenizer = GPT2Tokenizer.from_pretrained(model_name)
model = GPT2LMHeadModel.from_pretrained(model_name)

# Move model to GPU if available
device = torch.device("cuda" if torch.cuda.is_available() else "cpu")
model.to(device)

# Step 4: Define the chatbot logic
def generate_response(prompt):
    # Encode the user's input
    input_ids = tokenizer.encode(prompt, return_tensors='pt').to(device)

    # Generate text until the output length (which includes the input length) reaches 50
    output = model.generate(input_ids, max_length=50, num_return_sequences=1, no_repeat_ngram_size=2)

    # Decode the generated tokens and return the response
    response = tokenizer.decode(output[0], skip_special_tokens=True)
    return response

# Step 5: Test and interact with the chatbot
if __name__ == "__main__":
    print("Chatbot: Hello! How can I assist you today?")
    while True:
        user_input = input("You: ")
        if user_input.lower() in ['exit', 'quit']:
            print("Chatbot: Goodbye!")
            break
        response = generate_response(user_input)
        print(f"Chatbot: {response}")
```

### Final Answer

To create a chatbot using a large language model like GPT-2, you first need to choose the appropriate model based on your requirements. Then, set up your Python environment with the necessary libraries, particularly the Hugging Face Transformers library. Load the pre-trained model and tokenizer using the `GPT2LMHeadModel` and `GPT2Tokenizer` classes, respectively. Define the chatbot's interaction logic in a function that takes user input, encodes it, generates a response using the model, and decodes the model's output back into text. Implement a simple loop to continuously interact with the chatbot, allowing users to input messages and receive responses from the chatbot. Test the chatbot thoroughly to ensure it meets your expectations and refine it based on feedback and performance. The provided Python code sets up a basic framework for achieving this, using GPT-2 as the example model, and can be extended or modified according to specific needs and the capabilities of other large language models.

Figure 18: How to create a chatbot using an LLM: the answer from Qwen-2.5-7B trained with the calibrated 85%-accurate RM.

**Question:** Generate python code for creating a chatbot using a large language model

**Answer:** To create a chatbot using a large language model in Python, we can leverage libraries such as `transformers` by Hugging Face, which provides easy access to various pre-trained language models including GPT-3 compatible models (like GPT-2 or larger models in the GPT series, BERT, etc.). Below is a step-by-step reasoning process followed by the Python code to create a simple chatbot:

### Reasoning Process:
1. **Choose a Language Model**: Select a pre-trained language model from the `transformers` library. For this example, we'll use `GPT-2` due to its balance between capability and ease of use.
2. **Import Necessary Libraries**: Import `transformers` and `torch` (for handling tensors and computations).
3. **Load the Model and Tokenizer**: Load the GPT-2 model and its associated tokenizer. The tokenizer is crucial for converting text inputs into tokens that the model can understand.
4. **Define the Chatbot Functionality**: Create a function that takes user input, tokenizes it, feeds it through the model to get a response, and then tokenizes and decodes the model's output back into text.
5. **Handle Conversation Context**: While simple, for this example, we won't maintain conversation context but in a production setting, you might want to feed the previous interaction history to the model to make the chat more coherent.

### Python Code:
```python
import torch
from transformers import GPT2Tokenizer, GPT2LMHeadModel

# Step 1: Load the GPT-2 model and tokenizer
model_name = 'gpt2'  # Using gpt2, can also use 'gpt2-medium', 'gpt2-large', 'gpt2-xl' for more capacity
model = GPT2LMHeadModel.from_pretrained(model_name)
tokenizer = GPT2Tokenizer.from_pretrained(model_name)

# Ensure the model is set to evaluation mode and moved to the appropriate device (CPU or GPU)
model.eval()
if torch.cuda.is_available():
    model.to(torch.device("cuda"))

# Step 2: Define the chat function
def chat_with_bot(user_input):
    # Tokenize the user input
    input_ids = tokenizer.encode(user_input, return_tensors='pt')

    # Generate a response
    # Here we generate 100 tokens (you can adjust this based on expected response length)
    output = model.generate(input_ids, max_length=input_ids.shape[1] + 100, num_return_sequences=1)

    # Decode the response back to text
    bot_response = tokenizer.decode(output[0], skip_special_tokens=True)

    return bot_response

# Example usage
user_message = "Hello, how are you?"
response = chat_with_bot(user_message)
print("Bot says:", response)
```

### Summary of Reasoning and Answer:
In this solution, we first imported the necessary components from the `transformers` library, specifically the GPT-2 model and its tokenizer. We loaded these into our script ensuring they are ready for inference, with an option to use a GPU if available for speedup. The `chat_with_bot` function encapsulates the interaction: it tokenizes the user's input, uses the model to generate a response by extending the input sequence, and finally decodes the model's output back into human-readable text. This approach leverages the model's ability to generate text that continues from a given prompt, effectively simulating a chatbot response. Note that the quality and relevance of the responses depend heavily on the pre-training of the GPT-2 model and the specific tuning or fine-tuning it might have undergone for chatbot applications.

Figure 19: How to create a chatbot using an LLM: the answer from Qwen-2.5-7B trained with the original 85%-accurate RM.

**Question:** Generate python code for creating a chatbot using a large language model

**Answer:** To create a chatbot using a large language model in Python, we can follow these steps:

1. **Select a Language Model**: We'll use a pre-trained language model from Hugging Face's `transformers` library, such as `gpt-2` or `gpt-j`. For this example, we'll use `gpt-2`.

2. **Set Up the Environment**: Ensure you have Python installed along with the necessary libraries. You can install them using pip if they are not already installed:
    ```bash
    pip install transformers gradio
    ```

3. **Create the Chatbot**: We'll write a Python script that sets up the chatbot using the selected language model, handles user input, generates responses, and displays them in a user-friendly way using `gradio`.

Here's a complete Python script that accomplishes this:

```python
import gradio as gr
from transformers import pipeline

def generate_response(user_input):
    # Load the pre-trained model for text generation
    text_generator = pipeline('text-generation', model='gpt-2')

    # Generate a response using the model
    response = text_generator(user_input, max_length=150, num_return_sequences=1)

    # Extract and return the generated text
    return response[0]['generated_text']

# Set up the Gradio interface
iface = gr.Interface(
    fn=generate_response,  # The function to call when input changes
    inputs="text",         # Type of input (in this case, text)
    outputs="text",        # Type of output (text)
    title="GPT-2 Chatbot", # Title of the interface
    description="Talk to a chatbot powered by the GPT-2 model." # Description of the interface
)

# Launch the interface
iface.launch()
```

### Explanation of the Code:

- **Import Libraries**: We import `gradio` for creating the web interface and `pipeline` from `transformers` for easily accessing the language model.
- **Define Response Generation Function**: The `generate_response` function takes user input, uses the `gpt-2` model to generate a response, and returns the generated text.
- **Set Up Gradio Interface**: We define the Gradio interface using `gr.Interface`. We specify the function to be called (`generate_response`), the type of input (`text`), and the type of output (`text`). We also add a title and description to the interface.
- **Launch the Interface**: Finally, we call `iface.launch()` to start the web interface where users can interact with the chatbot.

This script sets up a basic chatbot using the GPT-2 model through a simple web interface. Users can type in questions or statements, and the chatbot will generate responses based on the model's predictions. This example can be extended and customized further based on specific requirements, such as integrating more sophisticated input handling, improving the model's parameters, or adding more advanced features to the chatbot.

Figure 20: How to create a chatbot using an LLM: the answer from Qwen-2.5-7B trained with the calibrated 65%-accurate RM.

**Question:** Generate python code for creating a chatbot using a large language model

**Answer:** To create a chatbot using a large language model in Python, you can follow these steps. This example will use the Hugging Face Transformers library, which provides easy access to a wide range of pre-trained models. We'll use the `gpt-2` model for this demonstration, but you can replace it with any other model from the Hugging Face Model Hub that suits your needs.

### Step 1: Setting up the Environment
First, ensure you have Python installed along with the necessary libraries. You'll need `transformers` by Hugging Face and `torch` for handling the model computations. You can install them using pip if they are not already installed:

```bash
pip install transformers torch
```

### Step 2: Importing Libraries and Loading the Model
Import the required libraries and load the pre-trained model. Here's how you can do it:

```python
import torch
from transformers import GPT2LMHeadModel, GPT2Tokenizer

# Load pre-trained model and tokenizer
model_name = 'gpt2'  # You can change this to any other model name from the Hugging Face Model Hub
model = GPT2LMHeadModel.from_pretrained(model_name)
tokenizer = GPT2Tokenizer.from_pretrained(model_name)

# Set the model to evaluation mode
model.eval()
```

### Step 3: Creating a Basic Chat Function
Now, let's create a simple chat function. This function will take user input, tokenize it, generate a response using the model, de-tokenize the response, and print it out. Note that the quality of responses will depend heavily on the model used and might require fine-tuning for better performance on specific tasks or domains.

```python
def chat_with_model(user_input):
    # Tokenize the user input
    input_ids = tokenizer.encode(user_input + tokenizer.eos_token, return_tensors='pt')

    # Generate a response
    output = model.generate(input_ids, max_length=150, num_return_sequences=1, no_repeat_ngram_size=2)

    # Decode and print the response
    response = tokenizer.decode(output[0], skip_special_tokens=True)
    print("Chatbot:", response)

# Example usage
user_input = "Hello, how are you?"
chat_with_model(user_input)
```

### Summary of Reasoning:
1. **Environment Setup**: Ensure all required libraries are installed and accessible.
2. **Model Loading**: Use the Hugging Face Transformers library to load a pre-trained large language model (e.g., `gpt-2`). This step includes both the model weights and the tokenizer needed to convert text into a format the model can understand and vice versa.
3. **Model Configuration**: Set the model to evaluation mode to ensure it operates in a way that is suitable for generating responses rather than training.
4. **Chat Function**: Create a function that takes user input, processes it through the model to generate a response, and then presents that response. This involves tokenizing the input, generating text based on the model, and de-tokenizing the output to human-readable form.

This basic chatbot can be expanded by adding features such as context handling, more sophisticated response generation, or integration with other services for a more interactive experience.</answer>

Figure 21: How to create a chatbot using an LLM: the answer from Qwen-2.5-7B trained with the original 65%-accurate RM.

