# OpenReview forum: "The Climb Carves Wisdom Deeper Than the Summit: On the Importance of Reasoning Patterns"
_ICLR.cc/2026/Conference — ICLR 2026 Conference Withdrawn Submission_

### Official Review · Reviewer_1Lib · 2025-10-27

**Soundness:** 3
**Presentation:** 3
**Contribution:** 2
**Rating:** 4
**Confidence:** 5

**Summary:**

This paper focuses on the noisy reward problem faced by large language models (LLMs) in reinforcement learning (RL) training. It first injects noise into verifiable tasks such as mathematics and finds that LLMs are robust to reward noise. Then, it proposes the "Reasoning Pattern Reward (RPR)", which rewards models only based on key reasoning phrases. Later, RPR is used to calibrate noisy reward models in non-verifiable tasks, effectively improving the reasoning ability and evaluation performance of LLMs and enabling small models to achieve effective reasoning, providing a new perspective for LLMs training.

**Strengths:**

1. Breaking away from the traditional RL approach that relies on rewarding correct answers, this paper clarifies the value of the reasoning process itself and proposes the RPR mechanism. Without verifying the correctness of answers, the model can achieve performance close to that of training with clean rewards only through key reasoning phrases.

2. Covering Qwen and Llama model series, it verifies hypotheses in both verifiable and non-verifiable tasks, and also analyzes the impact of reward models with different accuracies, ensuring the reliability and generality of conclusions in different scenarios.

3. The proposed method of calibrating noisy reward models with RPR not only improves the performance of models but also lowers the threshold for small models to achieve effective reasoning. Moreover, it introduces minimal overhead, providing a feasible solution for the application of LLMs in noisy real-world environments.

**Weaknesses:**

1. The idea of "INSIGHTS VIA MANUALLY INTRODUCED NOISE IN VERIFIABLE REWARDS" has been discussed and mentioned in many works. Meanwhile, I think this part has little connection with the subsequent content. If it must be retained, the currently tested benchmarks are too few and relatively easy. Whether the same insight can be found on more difficult benchmarks needs further discussion.

2. The acquisition and optimization of the keyword list are not mentioned in the paper, which is crucial for RL training and the ability to extend this idea to more models and fields.

3. The experiment of "Comparing LLMs trained with RMs of various accuracies" is interesting, but it seems unnecessary to train with models of different accuracies. Whether artificial disturbance can replace this part of the results, I think the effects of the two are the same.

4. There are few intuitive numerical comparisons throughout the paper. At the same time, the table notes do not clearly state the evaluated benchmarks, which is very unfriendly for understanding the scattered experimental results.

**Questions:**

Please refer to the Weaknesses.

---

### Official Review · Reviewer_RkNw · 2025-10-31

**Soundness:** 2
**Presentation:** 1
**Contribution:** 2
**Rating:** 2
**Confidence:** 5

**Summary:**

The paper studies RL post-training for LLMs under noisy, non-verifiable rewards, points out reasoning patterns can suffice: even with substantial noise, a simple “Reasoning Pattern Reward” (RPR) that rewards key reasoning phrases—without checking correctness—matches the peak performance of clean, verified rewards on math/QA. Building on this, they use RPR to calibrate noisy reward models for open-ended tasks, which reduces false negatives, boosts win rates, and even enables small models to learn useful reasoning behaviors.

**Strengths:**

- The analysis on flipped reward for RLVR is interesting. And I did agree with the interpretation of the reasoning patterns.
- The reasoning-pattern reward (RPR) is lightweight and, when combined with a reward model, mitigates false negatives and improves evaluation performance across Qwen and Llama.

**Weaknesses:**

- While some fancy analyses are provided, the core idea of this work is a trivial trick that scales up the low reward if reasoning patterns exist. Though I don't want to see it lacks novelty, it indeed lacks enough investigation to show this method is helpful, especially section 5 is too short. We desire the model training approach to enhance performance, but not to present superficial 'reasoning patterns', but what evaluation in section 5 only compared the model trained with calibrated RM and original (self-trained) RM, not rigorously to other setups. And if reasoning patterns matter in model training, these model trained on open-ended tasks with calibrated RM should also gain fair advantage on verifiable tasks such as math and coding. Unfortunately, none of these experiments are studied.
- The calibrated reward can be hacked, with only a fixed, small set. WIth such a design, it is easy to train even a weaker model which can only output some words using the diverse patterns without providing informative content.
- Experiment setup needs justification.
  - Current top-1 model with heavy effort achieves only 0.84 score according to RewardBench. And the 0.85 ACC RM is easily achieved on training with helpsteer3 only. This means such proxy accuracy provided a limited indication for estimating 'noise', and can mislead the follow-up discussion.
  - Though it claims to address open-ended questions which are not verifiable, there are some well known benchmarks to measure the alignment for RLHF, such as ArenaHard/AlpacaEval/WildBench etc, The new developed benchmark lacks enough analysis to show it is comprehensive and a good proxy to measure alignment on open-ended tasks.
  - As the main contribution is to calibrate noise reward in low-score region, it lacks justification why only author trained RM was used in experiment. A more straightforward setup is to incorporate the publicly available RM in LLM training, such as those top ones from RewardBench leaderboard.
- The writing is really confusing and hard to follow. For example, it is confusing to include a long but useless approach as discussion from L396-410.

**Questions:**

- What is the exact training setup for the experiment in section 5?

---

### Official Review · Reviewer_ecox · 2025-11-01

**Soundness:** 1
**Presentation:** 3
**Contribution:** 2
**Rating:** 2
**Confidence:** 3

**Summary:**

This paper hypothesized that reasoning patterns are valuable for models to learn and improve performance. The authors proposed Reasoning pattern Reward (RPR), and used it to calibrate reward model for RL.

**Strengths:**

- The paper is well motivated.
- The authors used experiments to identify interesting findings and experimentally tested hypothesis.

**Weaknesses:**

- More explanation of RPR will help readers understanding this paper. For example, the high model performance in Section 3 can be attributed to reward shaping and reduced variance. Without ruling out these factors, it is less convincing to claim reasoning pattern really helped.
- I found Section 5 seems an after-fact discussion. If we know where and when RM goes wrong, direct compensation without even involving RPR will help. The challenge is how to identify when compensation is required, which is omitted in the discussion. A naive threshold based approach is used instead.
- The experiments use same verifier for training and testing, which may lead to reward hacking.
- OOD performance is not reported.

**Questions:**

Please see weaknesses.

---

### Official Review · Reviewer_VrYj · 2025-11-02

**Soundness:** 3
**Presentation:** 3
**Contribution:** 2
**Rating:** 4
**Confidence:** 3

**Summary:**

The paper explains the role of reasoning patterns, besides the answer, when working with verifiable rewards in RL training of LLMs. The paper introduces a setup where verifiable rewards are flipped, while an additional reward on the reasoning chain is proposed. The authors then quantify the effect on RL training for Llama and Qwen model families. They study this on MATH-500 and GPQA and HelpSteer3.

**Strengths:**

1. Robust analysis on Llama and Qwen on multiple benchmarks.
2. Quantifying the robustness to noise for verifiable reward setups, e.g. flipping 40% of rewards while still keeping top accuracy.
3. Proposal of a simple heuristic based reward for the reasoning part and sharing practical insights in a clearly written submission.

**Weaknesses:**

Some of the main weakness, in my opinion, would be:
1. Lack of stronger baseline study and referencing earlier results in the realm of reasoning rewards. Specifically, a simpler, yet strong baseline, would be to study the effect of checking only the presence of a <think> token, rather than numerous are other keyphrases. This would also connect with DeepSeek-R1 as baseline.

2. No strong differentiation between verbosity and reasoning quality. The length of the reasoning chain does not necessarily mean higher quality reasoning. Particularly, a study on MATH or GPQA, going further in-depth on relating accuracy with reasoning length, as well as reasoning errors, would be required.

3. Overall the RPR approach is quite heuristic in nature. It would be better to ground the results in the body of work surrounding process-level rewards with PPO, at minimum studying it with an RM trained for reasoning pattern quantification.

**Questions:**

Please refer to weaknesses.

---

### Note · Authors · 2025-11-13

**Comment:**

We thank the reviewers for their valuable suggestions, which we will use to improve the paper.

**Withdrawal Confirmation:**

I have read and agree with the venue's withdrawal policy on behalf of myself and my co-authors.